# Spik4lite: Refactoring Neuromorphic Sparsity for Efficient Spiking Neural Networks on Commodity Edge Devices

**Yongzhi She** [1]  **Qihua Zhou** [1]  **Yuhao Wang** [1]  **Yaodong Huang** [1]  **Laizhong Cui** [2][1]  **Jingcai Guo** [3]

## Abstract

Recently, the spiking neural networks (SNNs) have shown great promise in enhancing AI task performance by utilizing the brain-inspired and energy-efficient computational paradigm via the binary $(0/1)$ spikes. Modern SNNs, especially those based on transformers, often require FPGA accelerators or neuromorphic chips to enable spike-driven computations. However, this domain-specific hardware is not always accessible on commodity edge devices like NVIDIA Jetsons, which may degrade SNNs' energy efficiency due to massive computational waste on inactive "0" spikes and finally undermine the usage boundary. This limitation raises an interesting question: *is it possible to make SNNs edge-friendly and tame the computations mostly on active "1" spikes?* We present the answer *yes* and propose Spik4lite, which serves as a lightweight plug-and-play module to significantly improve SNN's performance between model accuracy and computational efficiency. The key is to refactor SNN's channel-wise neuromorphic sparsity by zeroing out low-efficiency channels while proactively compensating for the eliminated spikes. Different from prior methods mainly focusing on optimizing the theoretical synaptic operations, our design philosophy can evolve the SNNs into a physically compact manner, thus inherently saving more computational and energy costs. Extensive experiments based on real edge devices show that Spik4lite can be integrated into existing SNN baselines to further improve their accuracy-and-efficiency performance, guaranteeing the model accuracy while saving the computational and energy costs.

[1]College of Computer Science and Software Engineering, Shenzhen University. [2]Guangdong Laboratory of Artificial Intelligence and Digital Economy (SZ), Shenzhen 518107, China. [3]The Hong Kong Polytechnic University. Correspondence to: Qihua Zhou <qihuazhou@szu.edu.cn>, Laizhong Cui <cuilz@szu.edu.cn>.

*Proceedings of the 43ʳᵈ International Conference on Machine Learning*, Seoul, South Korea. PMLR 306, 2026. Copyright 2026 by the author(s).

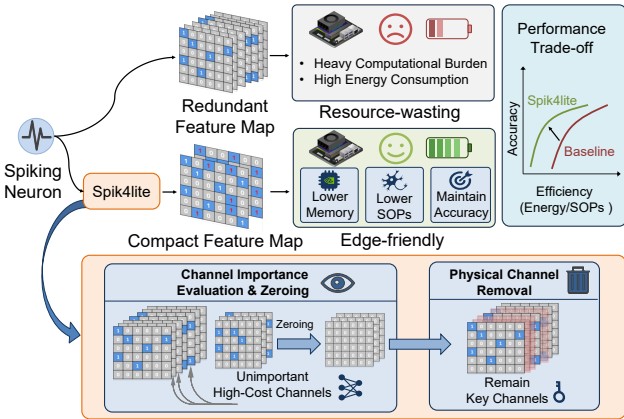

*Figure 1.* Overview of Spik4lite compression pipeline.

## 1. Introduction

Spiking Neural Networks (SNNs), recognized as the third generation of Artificial Neural Networks (ANNs) (Maass, 1997), have emerged as a promising paradigm for energy-efficient machine learning. By mimicking the biological brain's discrete, SNNs offer energy advantages on neuromorphic hardware (Roy et al., 2019; Davies et al., 2018) with event-driven computations. Direct training algorithms based on surrogate gradients (Wu et al., 2018) have emerged, and architectures have evolved from simple Convolutional Neural Networks (CNNs) to advanced SNN-Transformers like Spikingformer (Zhou et al., 2026) and Spike-driven Transformer (Yao et al., 2023). These SNNs can yield comparable performance as the modern ANNs when dealing with complex visual tasks (Deng et al., 2025). However, this better performance comes with a deeper network with more parameters, which often incur over-parameterization and heavy computational burdens, leading to a critical bottleneck for deploying deep SNNs on resource-constrained edge devices. To bridge this gap, many techniques of SNN compression have been explored. Specifically, these compression methods can be categorized into three main types: (1) Quantization, which compresses SNNs by reducing the numerical precision of synaptic weights and neuron states, such as membrane potentials (Wei et al., 2024). (2) Pruning, which compresses SNNs by removing redundant structures or activity-induced computations such as channels, kernels, synapses, or spikes (Li et al., 2024; Shi et al.,

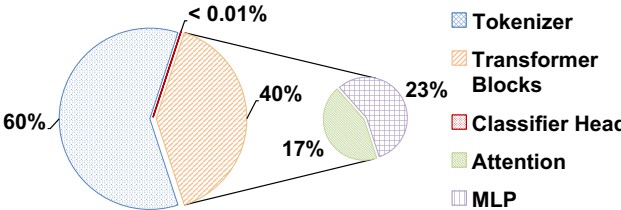

*Figure 2.* Spikingformer energy analysis on CIFAR10. The Tokenizer accounts for about 60% of total energy, followed by MLP (23%) and Attention (17%), while the Classifier Head consumes negligible energy ($< 0.01\%$).

2024a). (3) Neural Architecture Search (NAS), which automates the design of efficient SNNs by jointly optimizing network topology with spatial and temporal compression strategies (Liu et al., 2024). Most existing methods for designing lightweight SNNs are often based on CNN architectures, leaving the SNN-Transformers architecture largely unexplored. Furthermore, as illustrated in Figure 1, existing SNNs have not well explored the potential of commodity hardware, due to relying on specialized hardware for efficiency, yet wasting significant resources processing inactive "0" spikes on commodity hardware.

To pinpoint bottlenecks in SNN-Transformers, we analyzed the component-wise energy of Spikingformer (Figure 2). Considering that BN and linear layers are fused into convolutions during deployment (Ding et al., 2019; 2021; Hu et al., 2023; Chen et al., 2023), we focus on spike-based convolutions as the dominant energy source (Liang et al., 2022). Based on $45\,\mathrm{nm}$ CMOS standards ($E_{\mathrm{MAC}} = 4.6\,\mathrm{pJ}$, $E_{\mathrm{AC}} = 0.9\,\mathrm{pJ}$) (Horowitz, 2014), we observe that the Spiking Tokenizer and Feed-Forward Networks (MLPs) dominate consumption, with the Tokenizer alone accounting for about 60% of the total energy.

This identifies the compute-intensive operators within these modules as the primary targets. This observation motivates us to prioritize a "channel-wise pruning" strategy specifically for them. Aiming at this target, we propose a plug-and-play module called Spik4lite, which improves SNN-Transformers' efficiency on commodity hardware by leveraging channel-wise sparsity. Our key design is twofold. First, Spik4lite incorporates an energy-aware gating mechanism that penalizes channels with excessive Synaptic Operations (SOPs) during the training phase. Second, we physically remove these masked channels, which actually saves the computational and energy costs. This creates a physically compact model that guarantees real efficiency on commodity hardware while preserving the spike sparsity pattern of SNNs. Experiments show that our approach significantly reduces computational costs and achieves a better efficiency and accuracy trade-off over the baseline models. To the best of our knowledge, this is one of the first works to explore lightweight designs for SNN-Transformers on commodity hardware. Overall, our key contributions are as follows.

- **Bridging the hardware gap.** We extend the usage of deep SNNs from specialized neuromorphic chips to commodity devices, unleashing their potential on edge.

- **Real computation reduction.** We convert "spike sparsity" into "channel sparsity", ensuring that the hardware truly reduces calculations to be more efficient.

- **Efficient plug-and-play implementation.** We integrate Spik4lite into typical SNNs on commodity hardware, improving their accuracy-and-efficiency performance in an end-to-end manner.

## 2. Related Work

**Deep Spiking Neural Networks and Transformers.** With the maturation of surrogate gradient (SG) methods, SNNs have scaled from shallow networks to deep architectures like Spiking-ResNet (Zheng et al., 2021; Fang et al., 2021). Recently, SNN-Transformers have shown state-of-the-art (SOTA) performance in vision tasks. Zhou et al. (2023) first introduced the Spiking Self-Attention (SSA) mechanism to avoid multiplications in self-attention, while the Spike-driven Transformer (Yao et al., 2023; 2024) optimized key operations into spike-driven to achieve sparse-addition-only computation. Architectures like SpikingResformer (Shi et al., 2024b) and QKFormer (Zhou et al., 2024) further explored hybrid and hierarchical designs.

**Lightweight Spiking Neural Networks Designs.** To mitigate computational overhead, extensive research has been conducted on SNNs compression. Quantization methods like Q-SNNs (Wei et al., 2024) reduce memory footprint by lowering the bit width of synaptic weights and membrane potentials, while ReverB-SNN (Guo et al., 2025) reverses the bit allocation by using binary weights together with real-valued spike activations. Pruning is another widely used method for SNN compression. GradR (Chen et al., 2021) proposes a gradient-based rewiring mechanism to dynamically adjust connections. ESL-SNNs (Shen et al., 2023) proposes an evolutionary sparse training from scratch framework with dynamic pruning and regeneration to search for sparse connectivity, while Early-Time LTH (Kim et al., 2022) explores the Lottery Ticket Hypothesis and introduces early-time tickets that are identified with fewer timesteps to find sparse sub-networks. More recent pruning methods leverage dynamic SNN features. Activity Pruning (Bu et al., 2025) suppresses spike activity to reduce firing rates and SOPs. SCA (Li et al., 2024) uses spiking activity to guide structured channel or kernel pruning with dynamic pruning and regrowth. PQ Index (Shen et al., 2025) adapts rewiring ratios using a PQ index to guide pruning and regrowth. NAS methods like AutoSNN (Na et al., 2022) and SpikeDHS (Che et al., 2022) automate architecture search for light SNNs, while LitE-SNN (Liu et al., 2024) further

incorporates spatial and temporal compression choices into NAS, including pruning, mixed precision quantization, and timestep search. The readers can refer to the following papers to learn more about lightweight SNNs design (Deng et al., 2025; Qiu et al., 2025; Shi et al., 2024a). Existing works have made significant progress in lightweight SNN design, demonstrating that SNNs can be compressed into efficient models. However, many of these approaches are primarily optimized for neuromorphic hardware and may not be directly aligned with deployment on commodity edge devices. Here, our insight is to optimize the accuracy–efficiency trade-off not only on neuromorphic platforms but also on commodity hardware, which motivates us to propose the Spik4lite to expand the practical deployment boundary of deep SNNs on commodity edge devices.

## 3. Preliminary

**Spiking Neuron Model.** The spiking neuron mimics biological neurons by processing information through discrete spikes. Its mathematical model consists of three stages: charging, firing, and resetting. Specifically, at time step $t$, the neuron receives spatial input $X_t$. $X_t$ represents features extracted by deep learning operators such as Convolution, MLP, or Self-Attention. The dynamics of the LIF neuron are governed by an iterative process of charging, discharging, and resetting, which are formulated as:

$$
\begin{aligned}
\tilde{u}_t &= \beta u_{t-1} + X_t, \\
s_t &= \Theta(\tilde{u}_t - V_{th}), \\
u_t &= \tilde{u}_t \cdot (1 - s_t),
\end{aligned}
\tag{1}
$$

where $\tilde{u}_t$ denotes the membrane potential before firing, and $\beta \in [0, 1]$ is the decay factor controlling the leakage of temporal information $u_{t-1}$. The spike generation is determined by the Heaviside step function $\Theta(\cdot)$. Specifically, if the membrane potential exceeds the threshold $V_{th}$, the neuron fires a spike ($s_t = 1$) and resets $u_t$ to zero, but otherwise remains silent ($s_t = 0$) and accumulates the potential.

**The Gap Between SNNs and Commodity Hardware.** As shown in Eq. (1), the output $s_t$ is strictly binary. In deep SNNs, the firing activity is highly sparse, meaning $s_t$ equals 0 in most cases. Theoretically, these "0" events should consume no energy. However, on standard commodity hardware, these zeros still participate in dense matrix multiplications, causing resource waste. Since the spatial output $s_t$ determines the activation of subsequent channels, we leverage this natural spike sparsity to identify and remove redundant channels for computation reduction, thereby bridging the gap between SNNs and commodity hardware.

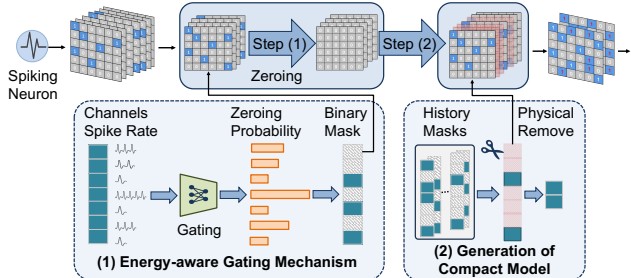

*Figure 3.* The detailed process of the Spik4lite module.

## 4. Methodology

### 4.1. Spik4lite Overview

According to our preliminary experiments (see Appendix A), static sparsity distributions cannot optimally balance the non-linear trade-off between energy and accuracy. To address this, we propose Spik4lite, a dynamic structural pruning framework for SNN-Transformers. As illustrated in Figure 3, the framework optimizes the network in two steps: (1) Energy-aware Gating Mechanism in Sections 4.2 and 4.3, and (2) Generation of Compact Model in Section 4.4.

### 4.2. Energy-aware Gating Mechanism

**Lightweight Gating with Energy Constraints.** In SNNs, information density is highly correlated with spiking frequency. Therefore, our gating mechanism takes the channel-wise firing rate vector as input. A lightweight linear layer projects this vector to produce unnormalized logits $\boldsymbol{\pi}_{l,c} \in \mathbb{R}^2$, which correspond to two states: *zeroing* (index 0) and *keep* (index 1). To prevent premature channel collapse during the early stages of training, the bias of linear is explicitly initialized to strongly favor the "keep" state (*e.g.,* setting the initial value for the keep index higher than the drop index). A critical challenge in our design is balancing task performance and energy consumption. High firing rates often contribute to higher accuracy but incur energy costs. Given an SNN denoted by $f(\cdot)$, parameterized by network weights $\theta$ and gating parameters $\phi$, we formulate the learning of the sparse SNN as a loss minimization problem with an energy penalty term:

$$
\underset{\theta, \phi}{\arg \min} \, \mathcal{L}_{\text{task}}(f(\cdot; \theta, \phi)) + \lambda \mathcal{L}_{\text{energy}}(f(\cdot; \theta, \phi)). \tag{2}
$$

Here, $\mathcal{L}_{\text{task}}$ represents the primary task loss (*e.g.,* Cross-Entropy). Note that $\mathcal{L}_{\text{energy}}$ is the energy-aware penalty term, which incorporates the dynamic SOPs and historical firing density of each layer (detailed in Section 4.3). The penalty coefficient $\lambda$ controls the trade-off. During backpropagation, $\mathcal{L}_{\text{energy}}$ exerts gradient pressure directly on $\phi$, penalizing the retention of computationally expensive channels unless they significantly contribute to minimizing $\mathcal{L}_{\text{task}}$.

**Differentiable Gating via Gumbel-Softmax.** The gating layer projects the input firing rate to the raw logit vector $\boldsymbol{\pi}_{l,c} = [\pi_{l,c,0}, \pi_{l,c,1}]$. To achieve efficient exploration of the pruning structure and stable optimization of the energy objective at the same time, we propose a decoupled strategy for mask generation and energy regularization. A naive application of standard Softmax for generating binary masks often leads to suboptimal pruning: once a channel is suppressed, its gradient vanishes, making it difficult to reactivate. To address this non-differentiability in discrete selection, we employ the Gumbel-Softmax relaxation (Jang et al., 2017) to facilitate *Stochastic Exploration*. The logits are perturbed with Gumbel noise $\mathbf{g} \sim \text{Gumbel}(0, 1)$ and scaled by a temperature $\tau$:

$$\tilde{\mathbf{P}}_{l,c} = \text{Softmax}\left(\frac{\boldsymbol{\pi}_{l,c} + \mathbf{g}}{\tau}\right) \in \mathbb{R}^2, \qquad (3)$$

where $\tau$ decays during training to transition from random exploration to deterministic selection. To ensure consistency between training and inference, specifically to prevent the distortion of spike timing caused by soft probability scaling, we utilize the "Hard" Gumbel-Softmax trick. The discrete binary mask $m_{l,c} \in \{0,1\}$ used in the forward pass is generated by discretizing the stochastic probability:

$$m_{l,c} = \mathbb{I}\left(\underset{k \in \{0,1\}}{\arg\max} \tilde{\mathbf{P}}_{l,c,k} = 1\right). \qquad (4)$$

Here, $\mathbb{I}(\cdot)$ is the indicator function, where $m_{l,c} = 1$ indicates the channel is retained. However, the discrete sampling in Eq. (4) is non-differentiable. To solve this, we apply the Straight-Through Estimator (STE) using the soft probability $\tilde{\mathbf{P}}_{l,c}$ as a continuous approximation. As $\tau \to 0$, $\tilde{\mathbf{P}}_{l,c}$ closely mimics the discrete mask. During backpropagation, we simply replace the gradient of the mask with that of the soft probability: $\partial m / \partial \boldsymbol{\pi} \approx \partial \tilde{\mathbf{P}} / \partial \boldsymbol{\pi}$. This allows gradients to flow through the sampling step to update $\boldsymbol{\pi}$, while keeping the binary mask in the forward pass. Although stochasticity helps escape local optima during mask generation, it introduces high variance into the energy penalty term, potentially destabilizing optimization. Therefore, we decouple the regularization path from the sampling path to ensure *Stable Regularization*. The energy penalty $\mathcal{L}_{\text{energy}}$ is computed using the standard, deterministic probability distribution without Gumbel noise or temperature scaling:

$$\mathbf{P}_{l,c} = \text{Softmax}(\boldsymbol{\pi}_{l,c}). \qquad (5)$$

The loss term minimizes the expected energy cost based on the probability of "Keep". Consequently, the gradient flow to the logits $\boldsymbol{\pi}$ is composed of two distinct components:

$$\frac{\partial \mathcal{L}_{\text{total}}}{\partial \boldsymbol{\pi}} = \underbrace{\frac{\partial \mathcal{L}_{\text{task}}}{\partial m} \cdot \frac{\partial \tilde{\mathbf{P}}}{\partial \boldsymbol{\pi}}}_{\text{Stochastic Exploration}} + \lambda \cdot \underbrace{\frac{\partial \mathcal{L}_{\text{energy}}}{\partial \mathbf{P}} \cdot \frac{\partial \mathbf{P}}{\partial \boldsymbol{\pi}}}_{\text{Stable Regularization}}. \qquad (6)$$

This formulation establishes a competition between two training objectives: the task loss drives the network to preserve performance-critical channels, while the energy loss explicitly forces the removal of computationally expensive features to minimize power consumption.

### 4.3. Refinement of Energy Constraints

**Formulation of Dynamic Energy Cost.** Since the firing rate of spiking neurons varies across time-steps and input batches, directly using the firing rate from a single batch will lead to unstable gradients for the gating parameters during learning. To solve this, we use a temporal smoothing strategy to estimate the long-term firing statistics. Let $r_{l,c}^{(k)}$ be the firing rate of the $c$-th input channel in layer $l$ at iteration $k$. We use an Exponential Moving Average (EMA) to track the long-term activity:

$$\hat{r}_{l,c}^k = \mu \cdot \hat{r}_{l,c}^{k-1} + (1 - \mu) \cdot r_{l,c}^k, \quad \mu \in [0,1], \qquad (7)$$

where $\mu$ is a momentum coefficient. This smoothed value $\hat{r}_{l,c}$ reliably reflects the historical firing rate of the channel. To effectively guide pruning for standard edge devices, we need a cost metric that aligns with the inherent properties of SNNs. A naive approach is to only penalize the FLOPs of channels, which represent the static computational cost based on the network structure. However, this metric ignores spike dynamics. If we prune based strictly on static FLOPs, the network tends to retain a minimal set of channels but forces them to fire continuously to preserve information. This causes the SNN to degenerate into a dense, high-frequency network, undermining the efficiency gains of spike sparsity. To avoid this, we incorporate the firing rate $\hat{r}_{l,c}$ into our penalty. We calculate the SOPs to represent the effective dynamic load. The SOPs for the $c$-th channel in the $l$-th layer is defined as:

$$\text{SOPs}_{(l,c)} = \text{FLOPs}_{(l,c)} \times T \times \hat{r}_{l,c}, \qquad (8)$$

where $\text{FLOPs}_{(l,c)}$ is the computational cost determined by the kernel size and feature map size, $\hat{r}_{l,c}$ is the average firing rate of the input spike train, and $T$ is the number of time steps of the spiking neuron. Unlike standard FLOPs, which treat all neurons as equally active, SOPs quantify the event-driven workload. This SOPs-guided penalty acts as a strict regularizer, suppressing high-activity channels that consume excessive energy but yield low marginal gains. Once a channel is consistently masked due to its high SOPs cost, it is physically removed from the network structure through an iterative consolidation process (detailed in Section 4.4). This achieves FLOPs reduction on standard hardware while maintaining the sparse, event-driven nature of the SNN. Consequently, optimizing theoretical SOPs can effectively guide the model to a physically compact structure.

**Layer-wise Normalized Energy Loss.** Finally, we address the issue of scale imbalance. In a deep network, differ-

ent layers have different energy magnitudes due to varying feature map sizes and channel counts. If we directly minimize the total sum of SOPs, the optimization process will be biased. It will focus primarily on the early layers with large feature maps while neglecting the deeper layers. To solve this, we employ a layer-wise normalized energy loss. Instead of using raw values, we normalize the energy consumption of each layer by its total cost when all channels are kept. The final energy loss is defined as the average of these normalized ratios across all learnable layers:

$$\mathcal{L}_{\text{energy}} = \frac{1}{N} \sum_{l=1}^{N} \left( \frac{\sum_c P_{l,c} \cdot \text{SOPs}_{(l,c)}}{\sum_c \text{SOPs}_{(l,c)}} \right), \qquad (9)$$

where $N$ is the total number of layers and $P_{l,c}$ denotes the probability of keeping the $c$-th channel. This normalization prevents instability caused by the SOPs in different layers. This operation makes the optimizer focus on the relative sparsity of each layer to remove unimportant channels, thus eliminating redundancy uniformly.

**Joint Optimization Objective.** Combining the task performance objective with the energy constraint regularization, we formulate the learning of the sparse SNN as a joint optimization problem. The goal is to find the optimal network weights $\theta$ and gating logits $\boldsymbol{\pi}$ that minimize the task error while strictly constraining the energy consumption. Unlike the forward pass, which utilizes stochastic sampling for exploration, the energy penalty is computed via a deterministic path to ensure gradient stability. By substituting the specific definition of the layer-wise normalized energy cost into the general objective, the final consolidated minimization problem is formulated as:

$$\arg\min_{\theta, \boldsymbol{\pi}} \quad \underbrace{\mathcal{L}_{\text{task}} \left( f(\mathbf{x}; \theta, \mathbf{m}(\boldsymbol{\pi}, \mathbf{g}, \tau)) \right)}_{\text{Performance Objective}}$$
$$+ \lambda \underbrace{\frac{1}{N} \sum_{l=1}^{N} \left( \frac{\sum_c [\text{Softmax}(\boldsymbol{\pi}_{l,c})]_1 \cdot \text{SOPs}_{(l,c)}}{\sum_c \text{SOPs}_{(l,c)}} \right)}_{\text{Normalized Energy Constraint}}.$$

$$(10)$$

For the *Performance Objective*, the first term uses $\mathbf{m}(\cdot)$ to denote the binary mask generated via the Hard Gumbel-Softmax trick (Eq. (4)), which involves the Gumbel noise $\mathbf{g}$ and temperature $\tau$. This term ensures the subset of channels selected is discriminative for the target task. Regarding the *Normalized Energy Constraint*, this second term represents the expanded energy penalty. Here, $[\text{Softmax}(\boldsymbol{\pi}_{l,c})]_1$ extracts the probability of the "keep" state (index 1) from the raw logits in a deterministic manner. This probability acts as a soft scaling factor for the estimated dynamic energy SOPs, which is derived from the smoothed historical firing rates (Eq. (7)). The normalization term $\sum \text{SOPs}$ in the denominator ensures that the penalty is scale-invariant across layers, preventing the optimization from being dominated by shallow layers with large feature maps. The penalty coefficient $\lambda$ governs the trade-off strength: a larger $\lambda$ forces the logits $\boldsymbol{\pi}$ to evolve towards the "zeroing" state for high-cost channels, effectively pruning them from the network.

### 4.4. Generation of Compact Model

**Physically Remove Redundant Channels.** Due to the stochastic nature of Gumbel-Softmax sampling, relying on a single iteration for pruning decisions entails high variance. To ensure robustness, we adopt a cyclic pruning schedule with temporal accumulation. The training process is divided into accumulation intervals of $K$ epochs (*e.g.,* $K = 20$). Within each interval, we accumulate the history binary masks from the Gumbel-Softmax sampler to compute the average retention probability of each channel. At the end of each interval, channels with average retention probability lower than the threshold (*e.g., threshold* $= 0.5$) are identified as redundancy candidates. To prevent layer collapse and ensure training stability, we enforce a maximum pruning rate (*e.g.,* 5%–20% of current channels) per iteration. The channels with the lowest average retention probability are physically removed first, up to this limit. This process stops at a predefined epoch (*e.g.,* 30 epochs before completion) to allow for fine-tuning. Specifically, we resize the weight tensors by removing the corresponding rows or columns. To ensure smooth convergence, we also adjust the optimizer by slicing the stored states (like momentum buffers) to match the new weights, thereby preserving the training history of the remaining parameters.

**Globally Align Network Dimensions.** Unlike standard CNNs, SNN-Transformers contain residual connections and Multi-Head Self-Attention (MHSA) blocks that require strict topological constraints. This creates a strict dependency that removing a channel in one layer will affect the validity of the entire path. If we prune layers independently, the dimensions will mismatch at the addition points, breaking the network's flow. To address this, we apply different strategies based on the layer's connectivity. For internal layers isolated from the main residual stream, we prune channels based on local importance. For layers connected to the main data path, we treat all topologically coupled components as a unified group. We calculate a unified importance score by computing the element-wise arithmetic mean of the gating probabilities from all layers in the group. A global mask is generated to retain the top-ranked channels based on these aggregated scores. This mask is applied simultaneously to every component in the path, ensuring they shrink in sync. Finally, to satisfy the structural requirements of MHSA, we enforce the number of kept channels to be divisible by the number of attention heads.

**Permanently Strip Gating Modules.** In the final stage of training, the model has transformed into a compact network.

During the pruning phase, the gating routers acted as temporary scaffolds to identify unnecessary channels. Once the structure stabilizes (*e.g.,* in the last 30 epochs), these gating routers are no longer needed. Therefore, we permanently remove the gating layers and the energy penalty term, leaving a lightweight, static SNN. Finally, we fine-tune this compact model using only the task loss $\mathcal{L}_{task}$ to ensure a highly efficient model ready for deployment.

In Summary, the above three-stage procedure of Spik4lite training is described in Algorithm 1.

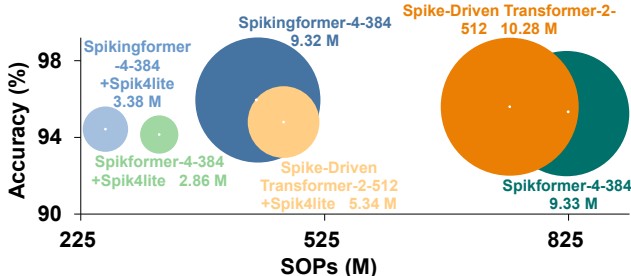

*Figure 4.* Comparative results of accuracy, SOPs, and parameters on CIFAR10 across different models.

---

**Algorithm 1** Training Framework of Spik4lite.

---

**Input:** Input data $X$, Labels $Y$.
**Output:** The lightweight SNN model.
1: Initialize network weights $\theta$ and gating logits $\boldsymbol{\pi}$; Initialize mask accumulator $\mathbf{m}_{history} = 0$;
2: **for** $i$ in $[1, epochs]$ **do**
3:     **Energy-aware Gating Mechanism:**
4:     Update smoothed firing rates $\hat{r}$ and sample binary masks $\mathbf{m}$ using Hard Gumbel-Softmax;
5:     Execute forward pass with $X_{masked} = X \odot \mathbf{m}$;
6:     **Refinement of Energy Constraints:**
7:     Compute gradients for task loss and layer-wise normalized energy penalty;
8:     Update $\theta$ and $\boldsymbol{\pi}$ during backpropagation with decoupled derivative paths;
9:     **Generation of Impact Model:**
10:     Accumulate binary masks $\mathbf{m}$ into $\mathbf{m}_{history}$ to track average retention probability;
11:     **if** pruning interval reached **then**
12:         Compute unified importance for coupled layers (Group-wise alignment);
13:         Identify redundant channels based on $\mathbf{m}_{history}$ and prune up to max rate;
14:         Physically remove channels, resize tensors, and slice optimizer states;
15:         Reset mask accumulator $\mathbf{m}_{history} = 0$;
16:     **end if**
17: **end for**
18: Permanently strip gating modules and fine-tune the compact model with $\mathcal{L}_{task}$ only;
19: **return:** The lightweight SNN.

---

## 5. Experiments

### 5.1. Experiment Settings

Our experiments are conducted using the SpikingJelly framework (Fang et al., 2023), an SNNs framework implemented based on PyTorch. We perform experiments on both static datasets CIFAR-10/100, and neuromorphic datasets DVS-CIFAR10 and DVS-128 Gesture. We apply our Spik4lite in following SNNs-Transformer models: Spikformer (Zhou et al., 2023), Spike-Driven Transformer (Yao et al., 2023) and Spikingformer (Zhou et al., 2026). Note that Spikformer and the more recent Spikingformer are two

distinct baselines, which employ different attention mechanisms. In our experiments, we use 1 NVIDIA-4090 GPU for all datasets during model training. We adjust the performance of model inference on NVIDIA Jetson Orin Nano 8 GB. Additional experimental details are in Appendix B.

### 5.2. End-to-end Performance

In this section, we present a comparison between standard SNN-Transformers and our Spik4lite-enhanced versions across different datasets. The results regarding accuracy, latency, memory footprint, parameters, SOPs and energy consumption are summarized in Table 1. We generally adhere to the experimental setups described in Zhou et al. (2026); Yao et al. (2023); Zhou et al. (2023), including network architecture and training configurations. Spik4lite demonstrates the performance improvements when integrated into different SNN-Transformers baselines. Our experiments indicate that our method achieves an optimal trade-off between efficiency and accuracy, as visualized in Figure 4. Specifically, the original network (*e.g.,* Spikingformer on CIFAR10), when enhanced by our method, retains only 30.7% of the original parameters (2.86 M vs. 9.32 M) yet achieves 1.58× increase in power efficiency (3.42 J vs. 5.42 J). It requires only 280.40 M SOPs for inference, with a negligible accuracy loss of 0.64%.

Performance on neuromorphic datasets is equally impressive. For instance, on the DVS-128 Gesture dataset, the SD-Transformer enhanced by Spik4lite reduces energy consumption by approximately 3.2× (9.24 J to 2.85 J) and slashes latency from 1.65 s to 0.46 s, all while maintaining competitive accuracy. The lower Peak Memory, exemplified by the reduction from 2614 MiB to 1613 MiB for Spikformer on CIFAR10, shows that our method works well on commodity edge devices with limited resources. At the same time, the drastically reduced SOPs across all benchmarks prove that it keeps the efficiency benefits for neuromorphic hardware. Therefore, our approach maintains the strengths of SNNs while extending them to commodity devices effectively.

*Table 1.* Results on different datasets. We compare the baselines with our Spik4lite-enhanced versions. "SD-Transformer" denotes Spike-Driven Transformer. Model-L-D represents a model with $L$ encoder blocks and an embedding dimension of $D$. We utilize four key metrics to evaluate model performance. **Peak Memory:** Represents the maximum memory allocated since the beginning of model inference. **Latency:** Measures the time required to process a single batch during inference. **SOPs:** During training, we estimate the SOPs using $\hat{r}_{l,c}$ to compute the energy penalty. For the evaluation, the reported SOPs are calculated based on the actual spike counts over the entire test dataset. This metric is widely used to estimate theoretical energy costs of SNN inference based on neuromorphic hardware. **Energy:** Represents the real-world average energy consumption required to complete a single batch. Unlike the theoretical SOPs, this data is monitored by the NVIDIA Jetson Power GUI during inference.

| Dataset | Architecture | Batch Size | T | Top-1 Acc.(%) | Latency (s) | Peak Mem. (MiB) | Param. (M) | SOPs (M) | Energy (J) |
|---|---|---|---|---|---|---|---|---|---|
| CIFAR10 | Spikformer-4-384 | 128 | 4 | 95.24 | 0.97 | 2614 | 9.33 | 822.64 | 5.34 |
| | + Spik4lite | 128 | 4 | 94.15 | **0.55** | **1613** | **2.85** | **321.49** | **3.08** |
| | SD-Transformer-2-512 | 128 | 4 | 95.41 | 0.92 | 2654 | 10.28 | 752.62 | 5.34 |
| | + Spik4lite | 128 | 4 | 94.78 | **0.65** | **2243** | **5.34** | **474.47** | **3.64** |
| | Spikingformer-4-384 | 128 | 4 | 95.07 | 0.95 | 2643 | 9.32 | 442.85 | 5.42 |
| | + Spik4lite | 128 | 4 | 94.43 | **0.61** | **1817** | **2.86** | **280.40** | **3.42** |
| CIFAR100 | Spikformer-4-384 | 128 | 4 | 78.0 | 1.04 | 2615 | 9.36 | 1038.24 | 5.72 |
| | + Spik4lite | 128 | 4 | 74.19 | **0.56** | **1840** | **2.54** | **366.80** | **3.08** |
| | SD-Transformer-2-512 | 128 | 4 | 78.97 | 0.88 | 2655 | 10.28 | 835.88 | 5.11 |
| | + Spik4lite | 128 | 4 | 76.25 | **0.79** | **2526** | **6.06** | **601.78** | **4.43** |
| | Spikingformer-4-384 | 128 | 4 | 79.07 | 1.68 | 2644 | 9.32 | 533.71 | 9.74 |
| | + Spik4lite | 128 | 4 | 77.07 | **0.72** | **1893** | **5.05** | **378.66** | **3.96** |
| DVS-CIFAR10 | Spikformer-4-384 | 16 | 16 | 79.9 | 0.54 | 2725 | 2.59 | 850.78 | 3.4 |
| | + Spik4lite | 16 | 16 | 76.4 | **0.51** | **2702** | **1.73** | **498.26** | **2.81** |
| | SD-Transformer-2-512 | 16 | 16 | 79.3 | 0.94 | 2623 | 2.57 | 2302.39 | 5.08 |
| | + Spik4lite | 16 | 16 | 78.1 | **0.71** | **1970** | **1.44** | **906.02** | **3.91** |
| | Spikingformer-4-384 | 16 | 16 | 80.7 | 0.55 | 2725 | 2.57 | 745.47 | 3.52 |
| | + Spik4lite | 16 | 16 | 77.1 | **0.53** | **2703** | **1.79** | **529.05** | **3.34** |
| DVS-128 Gesture | Spikformer-4-384 | 16 | 16 | 96.88 | 0.71 | 2725 | 2.59 | 554.83 | 4.33 |
| | + Spik4lite | 16 | 16 | 94.1 | **0.66** | **2699** | **0.65** | **207.71** | **4.09** |
| | SD-Transformer-2-512 | 14 | 16 | 97.56 | 1.65 | 2997 | 2.57 | 2551.85 | 9.24 |
| | + Spik4lite | 14 | 16 | 94.1 | **0.46** | **2390** | **0.63** | **170.21** | **2.85** |
| | Spikingformer-4-384 | 16 | 16 | 98.26 | 0.63 | 2725 | 2.57 | 535.78 | 3.84 |
| | + Spik4lite | 16 | 16 | 94.79 | **0.59** | **2701** | **1.11** | **438.85** | **3.54** |

## 5.3. Comparison of On-device Energy Efficiency

To evaluate the efficiency on real-world hardware, we measured power consumption on an NVIDIA Jetson Orin Nano 8 GB during inference for both the baseline and our improved model. Figure 5 illustrates the power data monitoring by NVIDIA Jetson Power GUI, which shows the power profiles over time. More results are provided in Appendix C.1.

**Latency Reduction.** As shown in Figure 5a, both models reach a peak power consumption of approximately 12 W during the active inference phase. However, the active inference duration for the baseline Spikingformer is approximately 130 s (14 s–144 s), while our improved model completes the same task in roughly 57 s (14 s–71 s), leading to a 2.3× speedup. Specifically, as to the DVS-128 Gesture task (Figure 5b), the improved SDT model reduces the inference latency from 35 s to just 10 s, achieving a 3.5× speedup.

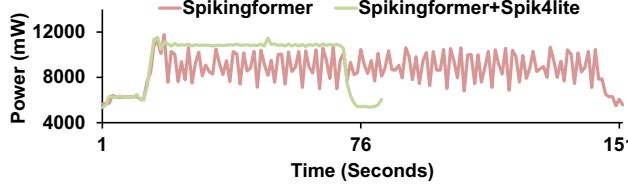

*(a)* Spikingformer power profile on CIFAR-100.

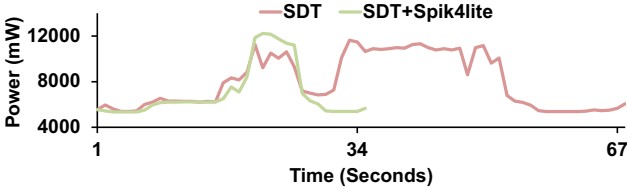

*(b)* SDT power profile on DVS-128 Gesture.

*Figure 5.* Real-time power consumption profiles on NVIDIA Jetson Orin Nano during a full inference pass over the entire test set. "SDT" denotes Spike-Driven Transformer. Note that while the figure logs data in milliwatts (mW) directly from the sensor, our analysis discusses efficiency in terms of watts (W).

**Accumulative Energy Cost.** Since total energy consumption corresponds to the integral of power over time ($E = \int_0^T P(t)\,dt$), the reduction of inference latency leads to lower energy costs per task. Despite exhibiting a similar peak power wattage, the improved model effectively reduces the total energy required by more than half, making it suitable for battery-constrained edge applications.

**Stability.** The proposed method also demonstrates superior power stability. As highlighted in Figure 5a, the baseline model exhibits severe power oscillations (fluctuating erratically between 7 W and 12 W) throughout the entire inference process. In contrast, our improved method maintains a consistent and stable power profile at the peak load ($\sim 10.8$ W), effectively eliminating the erratic power spikes and ensuring smoother hardware operation.

### 5.4. Abaltion Study

**Structure Evolution Analysis.** To better understand how Spik4lite autonomously optimizes the network architecture, we visualize the layer-wise channel evolution after training. As shown in Figure 6, the curves represent the number of retained channels in each layer of the Spikingformer on the CIFAR10 dataset. We observe a distinct pattern: the network learns to prune aggressively in the deep MLP blocks while preserving more capacity in the shallow, high-resolution Tokenizer layers initially. Specifically, deeper layers (*e.g.,* Layers 6, 8, 10, and 12) see a sharp reduction in channels, implying they contain significant redundancy. For instance, with $\lambda = 0.01$, the channel count in Layer 12 drops from the baseline of 1536 to just 96. In contrast, early layers (such as Layers 1-4) exhibit minor fluctuations before stabilizing, indicating their critical role in low-level feature extraction. This dynamic adjustment suggests that Spik4lite does not rely on fixed heuristics but adaptively redistributes the channel budget across layers, concentrating capacity where it is most beneficial for the task and converging to a compact structure.

**Ablation Study of Penalty Coefficient.** To investigate the trade-off between energy efficiency and performance, we performed an ablation study by varying the penalty coefficient $\lambda$. As illustrated in Figure 7, increasing $\lambda$ leads to a higher ratio of pruned channels, which significantly reduces computational cost while impacting accuracy. As shown in Figure 7a, the model initially exhibits high robustness to pruning. With a pruning ratio of 32%, the Top-1 accuracy remains at 95.07%, matching the baseline performance exactly. When the pruning ratio is increased to 55%, the model still sustains a high accuracy of 94.4%, with a negligible drop from the baseline of 95.07%. This suggests that our method effectively identifies and removes redundant channels without damaging the model's representational capability. As the penalty becomes more aggressive,

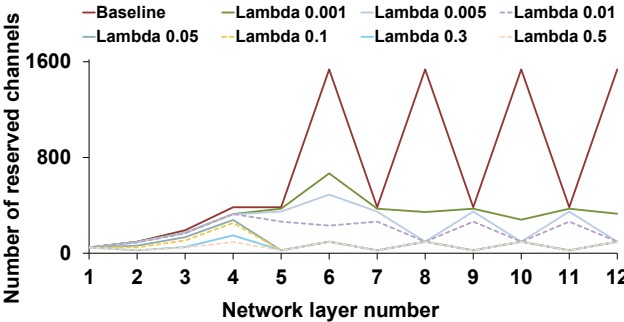

*Figure 6.* Network structures under various penalty coefficients.

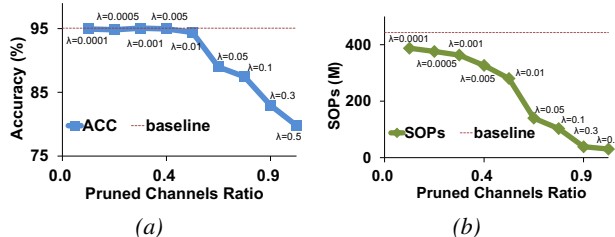

*Figure 7.* Accuracy and SOPs under different penalty coefficients.

a clearer trade-off emerges. At a high pruning ratio of 90%, the accuracy is preserved at 88.98%. Figure 7b highlights the corresponding gains in computational efficiency. The baseline model requires 442.85 M SOPs. However, as the pruned channel ratio increases, the SOPs decrease monotonically. In the "sweet spot" identified above (pruning ratios between 32% and 55%), the SOPs are significantly reduced to 363.09 M and 280.4 M, respectively, directly translating to lower energy consumption. Even at extreme sparsity levels (93%), where SOPs are minimized to merely 30.48 M and accuracy drops to 79.79%, the network avoids complete collapse. This resilience is attributed to the residual connections in the SNN-Transformer architecture. Even when channels in a layer are heavily pruned, the residual paths facilitate information propagation. These results demonstrate that our method offers a flexible balance, achieving substantial SOP reductions with controllable effects on accuracy.

## 6. Conclusion

In this paper, we propose Spik4lite to bridge the efficiency gap between SNNs and commodity hardware. By using a learnable gating, we identify and remove redundant channels based on their firing rates and energy costs. This results in a physically compact model that guarantees real reductions in computation and memory usage. Experiments on edge devices demonstrate that our method achieves a superior trade-off between accuracy and efficiency. This work provides a practical solution for deploying deep and efficient SNNs on resource-constrained edge devices.

## Software and Data

The source code, configuration files, and reproducibility instructions for Spik4lite are publicly available at https://github.com/yongzhishe/Spik4lite/.

## Acknowledgements

This work has been financially supported by the National Natural Science Foundation of China under Grant No. U23B2026, No. 24IAA01392 and No.62372305, Open Research Fund from Guangdong Laboratory of Artificial Intelligence and Digital Economy (SZ) under Grant No. GML-KF-26-13, Guangdong Basic and Applied Basic Research Foundation under Grant No. 2026A1515011556, No. 2024B1515040012, Shenzhen Science and Technology Program under Grant No. JCYJ20250604181612017, No. KJZD20230923114809020 and No. RCBS20231211090523043, Scientific Foundation for Youth Scholars of Shenzhen University under Grant No. RC20240254, No. RC20240032, and the Research Team Cultivation Program of Shenzhen University under Grant No. 2023QNT015.

## Impact Statement

This work studies edge-friendly spiking neural networks and reports findings that can help reduce computational and energy costs for on-device inference. Potential risks are similar to those of other vision models (*e.g.,* privacy concerns or biased performance), motivating responsible deployment and evaluation on diverse data.

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

## A. The Deceptive Nature of Global Sparsity

We present a pilot experiment to elucidate why existing rule-based pruning methods fail to achieve optimal efficiency on commodity hardware. This analysis is grounded in the premise that spike-based convolution is the major operation during the forward pass of LIF-based SNNs (Liang et al., 2022). Furthermore, the homogeneity of convolution allows the following BN and linear scaling transformation to be equivalently fused into the convolutional layer with an added bias during deployment, eliminating their separate energy costs (Ding et al., 2019; 2021; Hu et al., 2023; Chen et al., 2023). Specifically, we adopt the $45\,\mathrm{nm}$ CMOS technology standard for energy estimation, where the energy cost for a Multiply-and-Accumulate (MAC) operation is $E_{\mathrm{MAC}} = 4.6\,\mathrm{pJ}$, and for a spike-based Accumulate (AC) operation—corresponding to a Synaptic Operation (SOP)—is $E_{\mathrm{AC}} = 0.9\,\mathrm{pJ}$ (Horowitz, 2014). Following previous works, we postulate that *not all channels hold equal value regarding energy consumption and accuracy*. To validate this, we trained two variants of the Spikingformer-4-384 architecture. Both models were compressed to the same global channel sparsity of $\approx 84\%$, yet their layer-wise sparsity distributions were manually adjusted to distinct configurations: (1) Preserves more channels in the shallow, high-resolution tokenizer layers while aggressively pruning the deep MLP blocks. (2) Heavily prunes the shallow layers but retains more capacity in the deep layers. As shown in Table 2, despite having the identical parameter count (Global Sparsity $\approx 84\%$), the performance gap is stark. Configuration A achieves a higher Top-1 Accuracy (94.76%) but consumes significantly more energy ($0.37\,\mathrm{mJ}$) because it retains the computationally expensive shallow layers. In contrast, Configuration B reduces energy by 19% ($0.299\,\mathrm{mJ}$) by pruning the heavy shallow layers, but suffers a distinct accuracy drop (94.49%).

*Table 2.* Comparison of two pruning configurations with identical global sparsity ($\approx 84\%$) on Spikingformer-4-384. Tokenizer refers to the initial high-resolution layers, while MLP refers to the deeper blocks. Note how sparsity distribution dictates the trade-off between Accuracy and Energy.

| Metric | Configuration A | | Configuration B | |
|---|---|---|---|---|
| **Global Sparsity** | $\approx 84\%$ (7057/8400) | | $\approx 84\%$ (7093/8400) | |
| **Top-1 Accuracy** | **94.76%** | | 94.49% | |
| **SOPs Energy** | $0.370\,\mathrm{mJ}$ | | $0.299\,\mathrm{mJ}$ | |
| Layer-wise Retention | Ratio | Channels | Ratio | Channels |
| Tokenizer (Block 1-2) | **100%** | Full | $\sim 68\%$ | Pruned |
| Tokenizer (Block 3-4) | **81%-90%** | High | $\sim 68\%$ | Low |
| Deep MLP (Block 0) | $\sim 27\%$ | Low | $\sim 30\%$ | Medium |
| Deep MLP (Block 1-3) | $\sim 8\%$ | Very Low | $\sim 5\%$ | Ultra Low |

This observation reveals two critical insights:

**Global Sparsity is Deceptive.** Merely setting a target global sparsity (*e.g.,* 84%) is insufficient. The *distribution* of that sparsity determines the model's final performance.

**The Efficiency-Accuracy Trade-off is Non-Linear.** High-resolution shallow layers (like tokenizer) are "expensive" in terms of FLOPs but critical for feature extraction, manually crafting a rule to balance this trade-off is intractable.

Therefore, we argue that to bridge the gap between static inflexibility and optimal efficiency, the pruning policy must not be manually defined. Instead, it should be learned end-to-end, allowing the model to autonomously discover the Pareto-optimal structure that minimizes energy while maximizing accuracy.

## B. Experiment Details

### B.1. Experiment setup

Our experimental framework closely aligns with the methodology outlined in Spikformer (Zhou et al., 2023), Spike-Driven Transformer (Yao et al., 2023) and Spikingformer (Zhou et al., 2026). All of Baseline's code is publicly available, and we abide by their credentials. For four datasets, we adapt the Spik4lite to a variety of baseline models, following the precedents set by (Zhou et al., 2023; Yao et al., 2023; Zhou et al., 2026). For CIFAR, we maintain a timestep count of $T = 4$. For the neuromorphic datasets, we increase this to $T = 16$, respectively. Our experimental setup is consistent with all the SNN-Transformers baselines as follows.

**Spikformer with Spik4lite.** The training epoch is set to 410 for CIFAR10/100, 202 for DVS128 Gesture, and 106 for CIFAR10-DVS. The batch size is 128 for CIFAR10/100, 16 for DVS128 Gesture and CIFAR10-DVS. The learning rate is initialized to 0.0005 for CIFAR10/100, 0.001 for DVS128 Gesture and CIFAR10-DVS. All of them are reduced with cosine decay. The AdamW optimizer is employed for the training process on all datasets. We follow (Yao et al., 2023) to apply data augmentation on DVS128 Gesture and CIFAR10-DVS. In addition, the network structures used in CIFAR-10/100, CIFAR10-DVS, and DVS128 Gesture are Spikformer-4-384, Spikformer-2-256 and Spikformer-2-256.

**Spike Driven Transformer with Spik4lite.** The training epoch is set to 310 for CIFAR10/100 datasets, 210 for DVS128 Gesture/IFAR10-DVS. The batch size is 32 for CIFAR10/100, 16 for DVS128 Gesture and CIFAR10-DVS. The learning rate is initialized to 0.0005 for CIFAR10 and 0.0003 for CIFAR100, 0.001 for DVS128 Gesture and 0.01 for CIFAR10-DVS. Specifically, we utilize the LAMB optimizer for the CIFAR10-DVS dataset, while the AdamW optimizer is applied for the others. In addition, the network structures used in CIFAR-10/100, CIFAR10-DVS, and DVS128 Gesture are Spike-Driven Transformer-2-512, Spike-Driven Transformer-2-256 and Spike-Driven Transformer-2-256. Moreover, the batch size is set to 14 while inference for DVS128 Gesture, the original 16 will lead to an Out of Memory (OOM) problem on Jetson Orin Nano 8 GB.

**Spikingformer with Spik4lite.** The training epoch is set to 410 for CIFAR10/100 datasets, 202 for DVS128 Gesture, and 106 for CIFAR10-DVS. The batch size is 64 for CIFAR10/100, 16 for DVS128 Gesture and CIFAR10-DVS. The learning rate is initialized to 0.0005 for CIFAR10/100, 0.001 for DVS128 Gesture and CIFAR10-DVS. Consistent with the standard setting, the AdamW optimizer is adopted for all datasets. In addition, the network structures used in CIFAR-10/100, CIFAR10-DVS, and DVS128 Gesture are Spikingformer-4-384, Spikingformer-2-256 and Spikingformer-2-256.

*Table 3.* More experimental setup for different model architectures across various datasets. "SD-Transformer" denotes Spike-Driven Transformer.

| Dataset | Architecture | $\lambda$ | Pruning Interval | Pruning Threshold | Max Pruning Rate Embedding | Max Pruning Rate Mlp | Fine Tuning Epoch | Warmup Epoch |
|---|---|---|---|---|---|---|---|---|
| CIFAR10 | Spikformer* | 0.03 | 30 | 0.5 | 0.2 | 0.35 | 50 | 20 |
| | SD-Transformer* | 0.01 | 40 | 0.5 | 0.05 | 0.15 | 20 | 20 |
| | Spikingformer* | 0.01 | 70 | 0.5 | 0.3 | 0.5 | 50 | 20 |
| CIFAR100 | Spikformer* | 0.05 | 50 | 0.5 | 0.1 | 0.5 | 50 | 20 |
| | SD-Transformer* | 0.01 | 40 | 0.5 | 0.05 | 0.15 | 50 | 20 |
| | Spikingformer* | 0.005 | 60 | 0.2 | 0.1 | 0.15 | 80 | 20 |
| DVS-CIFAR10 | Spikformer* | 0.005 | 40 | 0.5 | 0.05 | 0.15 | 30 | 10 |
| | SD-Transformer* | 0.03 | 25 | 0.5 | 0.05 | 0.15 | 40 | 20 |
| | Spikingformer* | 0.01 | 20 | 0.5 | 0.02 | 0.25 | 30 | 10 |
| DVS-128 Gesture | Spikformer* | 0.03 | 30 | 0.5 | 0.05 | 0.25 | 30 | 10 |
| | SD-Transformer* | 0.05 | 20 | 0.5 | 0.1 | 0.2 | 30 | 10 |
| | Spikingformer* | 0.02 | 20 | 0.5 | 0.02 | 0.08 | 30 | 10 |

* Represents Spik4lite-enhanced models.

### B.2. Implementation Details

All training experiments are conducted on an NVIDIA GeForce RTX 4090 GPU (24 GB VRAM) and an Intel(R) Xeon(R) Gold 6430 CPU, running on Ubuntu 24.04.1 LTS. The software environment includes Python 3.12, PyTorch 2.5.0, and SpikingJelly 0.0.0.0.14. For edge deployment evaluation, we utilize the NVIDIA Jetson Orin Nano (8 GB) Developer Kit, running JetPack 6.2.0 (L4T 36.4.3). The inference latency and power consumption are measured using the onboard power sensors via the "jtop" utility (Jetson Stats) and the official NVIDIA Jetson Power GUI. To ensure a fair comparison for pure PyTorch-based inference, all our Jetson Orin Nano 8 GB experiments were conducted using active DVFS (15 W power mode) and PyTorch AMP (FP16), without TensorRT. It is worth noting that the Jetson platform utilizes a Unified Memory Architecture (UMA), where the 8 GB DRAM is shared between the CPU and GPU. Due to system overhead, the actual memory available for model inference is strictly less than the nominal capacity. In our experiments, the device maintains a baseline idle power consumption of approximately 5.5 W, with a system memory footprint of around 2.5 GB. The software environment is based on Python 3.10, while the versions of PyTorch and SpikingJelly are consistent with the training setup.

## C. Visualization

### C.1. Power Consumption Visualization

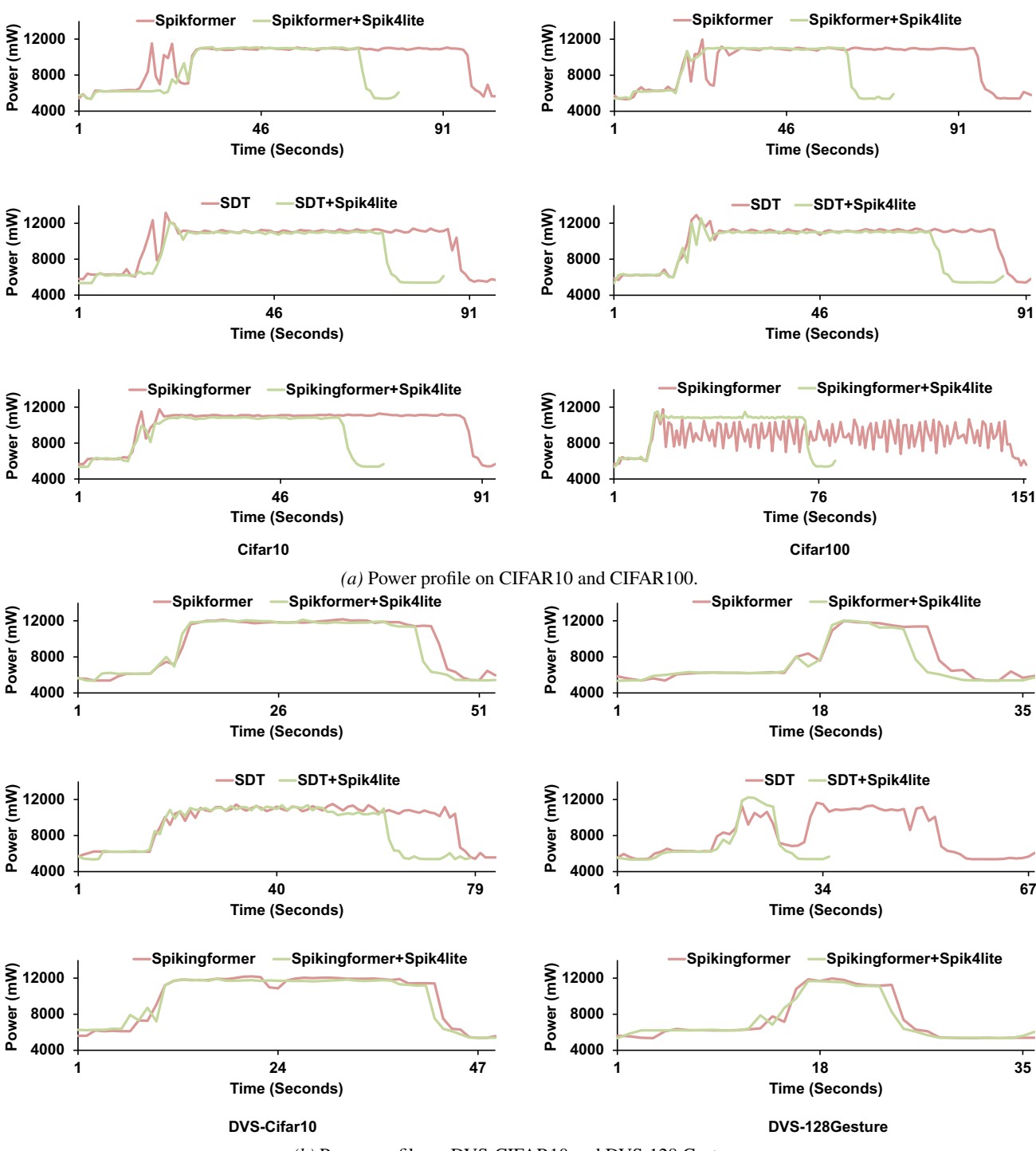

*(a)* Power profile on CIFAR10 and CIFAR100.

*(b)* Power profile on DVS-CIFAR10 and DVS-128 Gesture.

*Figure 8.* Power consumption profiles of different models during inference. "SDT" denotes Spike-Driven Transformer.

