# OpenReview forum: "Spik4lite: Refactoring Neuromorphic Sparsity for Efficient Spiking Neural Networks on Commodity Edge Devices"
_ICML.cc/2026/Conference — ICML 2026 regular_

### Official Review · Reviewer_VMZf · 2026-02-26

**Soundness:** 3
**Presentation:** 4
**Significance:** 4
**Originality:** 3
**Overall Recommendation:** 6
**Confidence:** 5

**Summary:**

The paper targets the “hardware gap” where SNN spike sparsity does not translate into real savings on commodity GPUs/edge devices because dense operators still process inactive spikes. It proposes Spik4lite, a plug-and-play module that refactors spike sparsity into channel-wise sparsity by learning energy-aware channel gates from firing-rate statistics, then physically removing low-efficiency channels to obtain a compact SNN-Transformer while aiming to preserve accuracy.

Methodologically, it uses Hard Gumbel-Softmax + STE for binary masks, with a decoupled deterministic path for the energy regularizer, and defines a SOPs-based dynamic cost using EMA-smoothed firing rates and layer-wise normalized energy loss; pruning is done cyclically with mask accumulation and topology-aware alignment for residual/MHSA constraints.

Experiments report latency/memory/SOPs reductions and on-device energy measured on Jetson Orin Nano, with moderate accuracy drops across CIFAR and DVS datasets.

**Compliance With Llm Reviewing Policy:**

Affirmed.

**Final Justification:**

This paper tackles an important yet underexplored problem in the SNN community: how to make SNNs truly practical on commodity edge devices. While much of the existing SNN literature focuses on algorithmic advances or theoretical efficiency gains, many methods still rely on specialized hardware and therefore remain difficult to deploy in real-world settings. In this regard, this work is highly valuable, as it explicitly addresses the gap between algorithmic design and practical edge deployment. I believe it can have a positive and beneficial impact on the SNN community by encouraging more research toward deployable, edge-friendly SNN systems.

**Key Questions For Authors:**

1. see "Weakness".

2. How sensitive are results to deployment regime (batch=1 vs batch>1, fixed clocks vs DVFS, TensorRT/FP16, etc.)? Please provide per-sample latency/energy and confidence intervals.

3. Can you add strong baselines: (a) FLOPs-based structured pruning, (b) activity-based SNN channel pruning, and (c) generic differentiable gating, all tuned for the same Jetson Orin Nano measurement protocol?

**Limitations:**

The approach requires retraining with gating + iterative pruning, which is expensive and hyperparameter-sensitive.

Generality to other edge platforms is uncertain.

**Strengths And Weaknesses:**

__Strengths__

- The paper cleanly motivates why spike sparsity fails on commodity hardware (energy breakdown + pruning target identification) and connects it to a concrete channel-pruning strategy with an explicit “physical removal” goal.

- The gating (Hard Gumbel-Softmax + STE), deterministic energy regularization path, SOPs formulation, and cyclic pruning/structure alignment are described with equations and an algorithmic template, which makes the intended workflow easy to follow.

__Weaknesses__

- Novelty is limited / incremental. The core technique resembles standard differentiable structured pruning: learnable binary gates (Gumbel-Softmax/STE) plus a regularizer, followed by iterative physical pruning. The “SNN-specific” contribution largely lies in using firing-rate-driven SOPs, but conceptually this is still a familiar pruning paradigm.

- Key claim “proactively compensating eliminated spikes” is underspecified. The abstract emphasizes compensation for removed spikes, but the main method description reads primarily as gating + pruning; it is unclear what concrete compensation mechanism is used (e.g., rescaling, distillation, residual reparameterization) beyond end-to-end training/fine-tuning.

- Experimental comparisons are incomplete for validating the claimed advantage. Results compare “baseline vs +Spik4lite” but do not convincingly separate whether gains come from (i) channel pruning itself vs (ii) the proposed SOPs-aware objective vs (iii) generic structured pruning baselines tuned for the same device/runtime constraints. Without strong competitive baselines on the same Jetson setup, “bridging the hardware gap” is not fully substantiated.

- On-device energy/latency protocol may not reflect real edge usage. Energy is reported as “average energy per batch” monitored via Jetson Power GUI, with large batch sizes for CIFAR (128) and smaller for DVS (16). Edge deployment often runs batch=1 with different kernel choices and memory behavior; the conclusions may shift significantly under batch-1 latency/energy and fixed-frequency vs DVFS-controlled settings (not detailed).

- Several knobs are described only with examples (e.g., pruning interval K, retention threshold, max pruning rate, temperature schedule), but not specified as final settings per experiment; moreover, topology-coupled pruning (group-wise alignment + “mean of gating probabilities”) is described qualitatively without enough implementation specificity to reproduce exactly for different SNN-Transformer blocks.

---

> ### Author Rebuttal · Authors · 2026-03-30
>
> We sincerely thank the reviewer for the constructive feedback and for acknowledging our work as technically solid with excellent presentation. We would like to address your concerns point by point below.
>
> **1. Novelty.**
> As detailed in our response to Reviewer fwxF ("1. Originality"), the primary scope of our work is to unleash the capability of SNNs on commodity devices. Our novelty lies in translating theoretical SNN sparsity into practical hardware efficiency on commodity edge devices, a capability lacking in previous methods.
>
> **2. Clarification on "Compensating".**
> The term "compensating" does not imply a standalone module. Instead, it is driven by our joint optimization objective (Eq. 10) and the fine-tuning phase (Section 4.4). Through differentiable soft-masking, the network updates the retained weights to recover the representational capacity of the suppressed channels during training.
>
> **3. Source of Gains (i) vs (ii) vs (iii).**
> The performance gains result from the integration of both (i)/(iii) and (ii). (i)/(iii) The structured channel pruning provides the actual efficiency for edge deployment. (ii) Meanwhile, our proposed SOPs-aware objective serves as the guidance for pruning while maintaining the natural spike sparsity of SNNs.
>
> **4. Clarification on Baselines (a), (b), and (c).**
> Some of the suggested approaches have been discussed in our work. (a) Section 4.3 explains that FLOPs-based structured pruning destroys the natural spike sparsity of SNNs and degenerates the model into a dense network. (b) As detailed in our response to Reviewer kwwT ("1. Baseline"), prior SNN pruning methods have mismatches in hardware adaptation and model architecture, so they are not our baseline. (c) The experiment in Appendix A demonstrates that generic differentiable gating will fall into sub-optimal sparsity.
>
> **5. Deployment Regime Details.** To clarify the experimental settings, all our Jetson Orin Nano 8G experiments were conducted using active DVFS (15W power mode) and PyTorch AMP (FP16), without TensorRT. More settings details are provided in Appendix B.2.
>
> **6. Per-Sample Metrics and Confidence Intervals.** We profiled the per-sample latency (Mean ± Std), 95% confidence intervals (CI), and energy across varying batch sizes (from 1 to 128) for Spikingformer-4-384 (T=4) on CIFAR-100. We present representative batch sizes (B=1, 8, 16, 128) below.
>
> | Batch Size | Architecture | Peak Mem. (MiB) | Latency/Samp. (ms) | 95% CI (ms) | Energy/Samp. (mJ) |
> | :---: | :--- | :---: | :---: | :---: | :---: |
> | 1 | Spikingformer-4-384 | 84 | 63.63 ± 4.42 | [63.55, 63.72] | 98.60 |
> | | + Spik4lite | 50 | 63.40 ± 3.86 | [63.32, 63.48] | 86.20 |
> | 8 | Spikingformer-4-384 | 221 | 7.96 ± 0.34 | [7.96, 7.97] | 44.00 |
> | | + Spik4lite | 155 | 7.96 ± 0.38 | [7.96, 7.97] | 36.20 |
> | 16 | Spikingformer-4-384 | 380 | 6.81 ± 0.24 | [6.80, 6.81] | 36.80 |
> | | + Spik4lite | 267 | 5.49 ± 0.12 | [5.49, 5.50] | 30.80 |
> | 128 | Spikingformer-4-384 | 2644 | 6.77 ± 2.01 | [6.73, 6.81] | 39.30 |
> | | + Spik4lite | 1893 | 5.14 ± 1.98 | [5.10, 5.18] | 28.30 |
>
> The conventional limitation on B=1 for edge deployment is a compromise due to the on-device overhead constraints. Our Spik4lite breaks this limitation by reducing physical footprints. Spik4lite not only performs well under B=1 but also outperforms the baseline under larger batch sizes (8-128), holding slower overhead growth than the baseline. Considering the performance trade-off displayed in the above table, we suggest setting B=8~32 for common edge usage and users can freely adjust this setting according to the overhead constraints in practice.
>
> **7. Details of hyperparameter settings and topology-coupled pruning.**
> Regarding the specific hyperparameter settings, we have comprehensively listed all parameters for every dataset and architecture in Appendix B.1 Table 3. To achieve topology-coupled pruning, we ensure structural consistency for linked components like residual or multi-head self-attention by averaging their accumulated gating probabilities. This unified score generates a single shared boolean mask applied together across layers, mathematically guaranteeing dimensional alignment. We will reorganize the demo code and release it on GitHub for easier use.
>
> **8. Training cost and hyperparameter sensitivity.**
> There might be a misunderstanding regarding the training overhead. Our method does not require an expensive separate retraining phase. The energy-aware gating, iterative pruning, and final static fine-tuning are integrated into an end-to-end run that uses the same epochs as the standard baseline. Furthermore, our approach is highly robust rather than hyperparameter-sensitive. As demonstrated in Appendix Table 3, we apply consistent hyperparameter settings such as retention thresholds and pruning intervals across different SNN architectures and datasets, without requiring tuning for each new task.

---

> > ### Author Rebuttal · Reviewer_VMZf · 2026-04-03
> >
> > Thank you for your response. It has addressed all of my concerns, and I am willing to raise my score to 6. I recommend acceptance of this paper.
> >
> > This paper tackles an important yet underexplored problem in the SNN community: how to make SNNs truly practical on commodity edge devices. While much of the existing SNN literature focuses on algorithmic advances or theoretical efficiency gains, many methods still rely on specialized hardware and therefore remain difficult to deploy in real-world settings. In this regard, this work is highly valuable, as it explicitly addresses the gap between algorithmic design and practical edge deployment. I believe it can have a positive and beneficial impact on the SNN community by encouraging more research toward deployable, edge-friendly SNN systems.

---

> > > ### Author Response · Authors · 2026-04-03
> > >
> > > Dear Reviewer VMZf,
> > >
> > > Thank you for your continued support. We are encouraged by your recognition of our work.
> > > Have a nice day!
> > >
> > > Thanks again,
> > > The authors

---

### Official Review · Reviewer_rATR · 2026-03-09

**Soundness:** 3
**Presentation:** 3
**Significance:** 2
**Originality:** 3
**Overall Recommendation:** 4
**Confidence:** 4

**Summary:**

This paper presents Spik4lite. It is a framework for Spiking Neural Networks (SNNs). The tool changes theoretical spike sparsity into physical channel sparsity. This helps standard edge devices like NVIDIA Jetsons run faster. These devices usually cannot handle fine-grained spikes efficiently. The method uses an energy-aware Gumbel-Softmax gate. It also uses a decoupled path to find and delete extra channels in SNN-Transformers. Tests show this lowers latency and power use. The goal is to run SNNs on common, cheap hardware.

**Compliance With Llm Reviewing Policy:**

Affirmed.

**Final Justification:**

the rebuttal addressed your main concerns so I would like to increase the score

**Key Questions For Authors:**

1.Your evaluation is limited to small datasets like CIFAR and DVS-128. Can you provide results on ImageNet-1K to prove that channel refactoring works for large-scale, high-dimensional tasks?

2.Comparison with Pruning Baselines: The paper only compares Spik4lite against unpruned models. How does your Gumbel-Softmax approach perform against standard SNN structured pruning methods?

**Limitations:**

1.	The evaluation relies on relatively small datasets (CIFAR and DVS-128). The effectiveness of this channel refactoring approach remains unverified on large-scale, high-dimensional tasks such as ImageNet-1K. Further testing is needed to confirm if the proposed pruning maintains accuracy as model complexity and task scale increase.

2.	The current evaluation compares Spik4lite only against unpruned models. It lacks direct comparisons with established structured pruning baselines adapted for SNNs. These comparisons are necessary to demonstrate the technical advantage of the Gumbel-Softmax approach. Without such results, the relative superiority of the proposed method remains unverified.

**Strengths And Weaknesses:**

Strengths

1.The framework converts spike sparsity into channel sparsity. This enables direct hardware acceleration on standard edge GPUs like NVIDIA Jetson. It avoids the need for specialized neuromorphic chips while achieving measurable power and latency reductions.

2.The method uses a decoupled regularization path within a Gumbel-Softmax gating system. It separates stochastic exploration from regularization to reduce gradient variance. This ensures stable convergence when learning discrete masks for channel selection.

3.The design enforces structural constraints specific to Spiking Transformers. It manages head-divisibility in multi-head attention and maintains alignment across residual connections. These constraints ensure the pruned model remains compatible with standard GPU kernels without sacrificing accuracy.

Weaknesses

1.The evaluation is limited to small datasets such as CIFAR and DVS-128. The absence of ImageNet-1K results is a significant limitation. It remains unproven whether channel refactoring is effective for large-scale, high-dimensional tasks.

2.The evaluation only compares Spik4lite against unpruned models. It lacks direct comparisons with established structured pruning baselines adapted for SNNs. Without these competitive results, the technical necessity and relative superiority of the proposed Gumbel-Softmax approach are not demonstrated.

---

> ### Author Rebuttal · Authors · 2026-03-30
>
> We sincerely thank you for the thorough review and for recognizing the originality and practical value of our work. We deeply appreciate your constructive feedback and would like to address your concerns point by point below.
>
> **1. ImageNet-1K experiments.**
> We take your suggestions and test our method on ImageNet-1K, using Spikingformer-4-384 ($T=4$) architecture. The results are shown below.
>
> | Architecture | Batch Size | T | Top-1 Acc.(%) | Latency (s) | Peak Mem. (MiB) | Param. (M) | SOPs (M) | Energy (J) |
> | :--- | :---: | :---: | :---: | :---: | :---: | :---: | :---: | :---: |
> | Spikingformer-4-384 | 16 | 4 | 69.96 | 0.56 | 2727 | 9.32 | 2398 | 3.25 |
> | + Spik4lite | 16 | 4 | 68.54 | 0.42 | 2305 | 6.76 | 1870 | 2.39 |
>
> Our preliminary results show that Spik4lite can achieve a significant reduction in latency, memory, and energy on ImageNet-1K, demonstrating our plug-and-play method can generalize to large-scale datasets, thus consistent with the performance mentioned in the main submission.
>
> **2. Clarification on baselines and our technical advantage.**
> Recall that our research objective is to unleash the capability of SNNs on commodity edge devices and remove the dependency on domain-specific hardware.
> Thus, prior lightweight SNN methods are not treated as our baselines due to two fundamental mismatches.
>
> - First, hardware adaptation. Existing methods optimize theoretical energy saving for neuromorphic chips, which cannot be directly applied to commodity edge devices. Conversely, our Spik4lite first targets real energy saving on commodity edge devices.
> - Second, model architecture. Existing methods primarily compress CNNs, which cannot deal with the strict topological dependencies in SNN-Transformers. Conversely, our Spik4lite target SOTA SNN-Transformers with complex topological dependencies.
>
> To address the concern regarding the technical advantage and necessity of our Gumbel-Softmax approach, we highlight the targeted validation provided in Appendix A, Table 2. This experiment demonstrates that under an identical global sparsity constraint, standard manual or rule-based pruning strategies inevitably result in either a severe accuracy drop or increased energy consumption. These results verify the relative superiority of our differentiable Gumbel-Softmax method, as it autonomously discovers the optimal sparsity distribution to achieve the best trade-off between task performance and actual edge-device efficiency.

---

### Official Review · Reviewer_fwxF · 2026-03-11

**Soundness:** 3
**Presentation:** 3
**Significance:** 2
**Originality:** 2
**Overall Recommendation:** 3
**Confidence:** 4

**Summary:**

This paper proposes Spik4lite, a pruning-and-compaction framework for SNN transformers on commodity edge devices, using firing-rate-based gating, an SOP-aware objective, and physical channel removal, with experiments on several SNN-transformer backbones including Jetson Orin Nano measurements.

**Compliance With Llm Reviewing Policy:**

Affirmed.

**Final Justification:**

The authors’ rebuttal provides a clear and targeted response to the concerns. However, overall, I believe the paper still has some limitations:
- Limited novelty: The response mainly emphasizes engineering value and system-level integration (e.g., bridging the hardware gap), but lacks a clear definition of concrete technical contributions.
- Insufficient comparative evaluation: Although additional ImageNet experiments are provided, the comparisons are still limited to internal ablations (with vs. without the proposed method), without systematic evaluation against other compression/pruning methods or stronger baselines.

Considering both the paper and the rebuttal, I will increase my score to 3.

**Key Questions For Authors:**

See weaknesses please.

**Limitations:**

See above.

**Strengths And Weaknesses:**

**Strengths**:

- The paper addresses a practically relevant setting: improving the deployment efficiency of SNN transformers on commodity edge hardware rather than only on specialized neuromorphic platforms.
- The method is end-to-end and technically coherent, combining gating, energy-aware pruning, physical channel removal, and compatibility handling for residual and multi-head attention structures.
- The evaluation covers multiple SNN-transformer backbones and datasets, and includes device-level measurements on Jetson in addition to accuracy, memory, latency, SOPs, and energy-related metrics.

**Weaknesses**:

- The main concern is originality: the overall pipeline combines familiar ingredients such as Hard Gumbel-Softmax gating, EMA-smoothed firing statistics, structured channel pruning, and post-pruning fine-tuning, so the distinctly new algorithmic component is not very clear.
- The empirical comparison is limited mainly to the original backbones and their Spik4lite-enhanced versions; it would be more convincing to compare against ANN models with similar parameter budgets, or ANN pruning/compression baselines that reach similar model size, under the same Jetson evaluation protocol.
- I would also like to see ImageNet-scale experiments, since the current evidence is limited to CIFAR-10/100 and neuromorphic datasets, while the paper positions the method as a general approach for SNN transformers.
- There appears to be a presentation inconsistency in Figure 4 for CIFAR10: the parameter annotations around the Spikingformer/Spikformer + Spik4lite bubbles are confusing, and they do not cleanly match the numbers reported in Table 1 (e.g., the table reports 2.86M parameters for Spikingformer-4-384 + Spik4lite on CIFAR10).

---

> ### Author Rebuttal · Authors · 2026-03-30
>
> We sincerely thank the reviewer for the thorough evaluation and constructive feedback. We are encouraged by your recognition of our method's practical relevance, and we would like to address your concerns point by point below.
>
> **1. Concerns regarding algorithmic originality.**
> We understand the reviewer's perspective that our method utilizes established techniques like Gumbel-Softmax and EMA, there might be a slight misunderstanding regarding our core contribution. Our work lies in deploying SNNs on commodity edge devices (e.g., NVIDIA Jetson), where utilizing pruning or compression techniques on neuromorphic hardware is a common focus of previous works. Different from solely proposing a new gating mathematical function, our originality lies in formulating a novel framework that translates theoretical SNN sparsity into practical hardware efficiency on commodity edge devices, which can not be achieved by previous methods. This preliminary drives our design philosophy, and our novelty lies as follows.
>
> - **Bridging the hardware gap.** First, we are not incrementally presenting a standard pruning trick by simply applying learnable gates (that is not our primary focus). Instead, we bridge the critical hardware gap to fundamentally improve the accuracy-and-efficiency trade-off of SNNs on non-neuromorphic chips, extending their practical usage boundary.
>
> - **Real computation reduction.** Second, Spik4lite converts "spike sparsity" into physical "channel sparsity," achieving actual computational and energy savings. By physically refactoring the network structure to eliminate redundant channels, our method directly translates theoretical SOP reductions into practical computation reduction, thereby guaranteeing optimal energy efficiency for edge deployment.
>
> - **Efficient plug-and-play implementation.** Third, our method is plug-and-play and compatible with common SNN-Transformers. We uniquely resolve the strict topological dependencies inherent in SNN-Transformers, greatly reducing memory footprint while preserving robust accuracy.
>
> **2. Clarification on the comparison against ANN.**
> We respectfully clarify that the inherent advantages of SNNs in low power and high efficiency on edge devices have been extensively validated by prior studies (e.g., Lite-SNN, Liu et al., IJCAI 2024). Consequently, the primary scope of our work is not to claim that SNNs outcompete ANNs. Instead, our primary scope is to unleash the capability of SNNs on commodity edge devices without relying on domain-specific hardware. Furthermore, to achieve efficient deployment and broaden the boundaries of SNN applications on commodity edge devices, a plug-and-play framework like Spik4lite is a must. Rather than designing a smaller SNN model from scratch, our method allows various existing SNN architectures to acquire immediate efficiency gains and improved hardware compatibility. This approach avoids the need to customize network structures for individual models, thereby significantly enhancing the universality and practical value of our method.
>
> **3. ImageNet-1K experiments.**
> We take your suggestions and test our method on ImageNet-1K, using Spikingformer-4-384 (T=4) architecture. The results are shown below.
>
> | Architecture | Batch Size | T | Top-1 Acc.(%) | Latency (s) | Peak Mem. (MiB) | Param. (M) | SOPs (M) | Energy (J) |
> | :--- | :---: | :---: | :---: | :---: | :---: | :---: | :---: | :---: |
> | Spikingformer-4-384 | 16 | 4 | 69.96 | 0.56 | 2727 | 9.32 | 2398 | 3.25 |
> | + Spik4lite | 16 | 4 | 68.54 | 0.42 | 2305 | 6.76 | 1870 | 2.39 |
>
> Our preliminary results show that Spik4lite can achieve a significant reduction in latency, memory, and energy on ImageNet-1K, demonstrating our plug-and-play method can generalize to large-scale datasets, thus consistent with the performance mentioned in the main submission.
>
> **4. Data presentation mismatch.**
> This is a minor oversight caused by rounding differences during plotting. The two values (2.85 and 2.86) of parameter count are from the same experiment. We will unify them to 2.86 in both Table 1 and Figure 4. We will also check the decimal precision across all figures and tables in the final version.

---

> > ### Author Rebuttal · Reviewer_fwxF · 2026-04-02
> >
> > Thank you for the detailed rebuttal. The authors have addressed most of my concerns, and I will increase my score to 3.

---

### Official Review · Reviewer_kwwT · 2026-03-13

**Soundness:** 2
**Presentation:** 3
**Significance:** 2
**Originality:** 2
**Overall Recommendation:** 4
**Confidence:** 3

**Summary:**

This paper proposes Spik4lite, a plug-and-play module for making spiking neural networks, especially SNN-Transformers, more efficient on commodity edge devices such as NVIDIA Jetson. The key idea is that although SNNs are theoretically sparse due to binary spikes, standard commodity hardware still wastes computation on inactive spikes because dense operators are executed anyway. To address this, the paper introduces an energy-aware channel gating mechanism that uses channel-wise firing rates and a normalized SOPs-based energy objective to identify expensive channels, then physically removes them during training to obtain a compact model. The paper evaluates the approach on multiple SNN-Transformer backbones and datasets, and reports improvements in latency, memory, parameters, SOPs, and measured on-device energy with relatively small accuracy drops.

**Compliance With Llm Reviewing Policy:**

Affirmed.

**Key Questions For Authors:**

[Written in weakness]

1. The empirical comparison is mostly against baseline architectures enhanced with Spik4lite, rather than a strong apples-to-apples comparison against the strongest prior lightweight SNN methods under fully matched settings. Will the other methods (except for Spikformer) can be with Spik4lite? This makes the practical gains promising, but somewhat harder to interpret as state-of-the-art evidence.

2. Would it be possible the proposed method can be compatible with other compression method for SNN? e.g., Quantization

3. The method is motivated specifically for SNN-Transformers on commodity hardware, but the broader generality remains somewhat unclear. For example, it would help to understand how well the method transfers across different normalization/operator choices.

**Limitations:**

Yes

**Strengths And Weaknesses:**

Strengths

- The paper addresses a clear and practically meaningful problem: the gap between the theoretical efficiency of sparse spike computation and the actual inefficiency of running SNNs on commodity hardware.

The paper evaluates the method on several SNN-Transformer architectures and includes real on-device measurements on NVIDIA Jetson Orin Nano, reporting not only SOPs but also latency, memory, and measured energy. This substantially strengthens the practical relevance of the work.

Weakness

- The empirical comparison is mostly against baseline architectures enhanced with Spik4lite, rather than a strong apples-to-apples comparison against the strongest prior lightweight SNN methods under fully matched settings. Will the other methods (except for Spikformer) can be with Spik4lite? This makes the practical gains promising, but somewhat harder to interpret as state-of-the-art evidence.
- Would it be possible the proposed method can be compatible with other compression method for SNN? e.g., Quantization
- The method is motivated specifically for SNN-Transformers on commodity hardware, but the broader generality remains somewhat unclear. For example, it would help to understand how well the method transfers across different normalization/operator choices.

---

> ### Author Rebuttal · Authors · 2026-03-30
>
> Thank you for your thoughtful review and for highlighting the practical significance of our work. We deeply appreciate your valuable suggestions and address your concerns point by point below.
>
> **1. Clarification on baselines and applicability to other architectures.**
> Recall that our research objective is to unleash the capability of SNNs on commodity edge devices and remove the dependency on domain-specific hardware.
> Thus, prior lightweight SNN methods are not treated as our baselines due to two fundamental mismatches.
> - First, hardware adaptation. Prior works optimize theoretical energy saving for neuromorphic chips, which cannot be directly applied to commodity edge devices. Conversely, our Spik4lite first targets real energy saving on commodity edge devices.
> - Second, model architecture. Prior works primarily compress CNNs, which cannot deal with the strict topological dependencies in SNN-Transformers. Conversely, our Spik4lite target SOTA SNN-Transformers with complex topological dependencies.
>
> Regarding applicability to other architectures, Spik4lite applies to various models beyond Spikformer, like Spike-Driven Transformer and Spikingformer. For instance, integrating it into the recent Spikingformer (Zhou et al., AAAI 2026) on CIFAR10 yields a 1.56× latency speedup and a 1.58× energy Saving, demonstrating its broad applicability and practical SOTA gains.
>
> **2. Compatibility with other compression methods.**
> Our method is compatible with other compression paradigms such as quantization. Spik4lite reduces architectural redundancy via physical channel removal, which can be seamlessly integrated with low-precision weights or membrane potentials for extreme edge efficiency.
>
> **3. Clarification on broader generality.**
> Spik4lite is designed as a highly adaptable plug-and-play module, ensuring broad generality. Our method is robust to various normalization schemes. Since standard operations like Batch Normalization or LayerNorm are equivalently fused into preceding spike-based layers during edge deployment (RepVGG, Ding et al., CVPR 2021), our channel-wise pruning remains agnostic to these normalization variants. Regarding operator choices, Spik4lite is natively compatible with common neural components like multi-dimensional convolutions and linear projections, as well as topological structures such as multi-head self-attention and residual connections.

---

> > ### Author Rebuttal · Reviewer_kwwT · 2026-04-02
> >
> > Thanks to the authors for resolving my concerns. Other concerns are resolved, but I still have concerns about "2. Compatibility with other compression methods." Do the authors have preliminary experiments on that? Because even two compression methods seem like orthogonal directions, they sometimes affect each other's performance.

---

> > > ### Author Response · Authors · 2026-04-04
> > >
> > > We sincerely thank the reviewer for the prompt feedback and the insightful follow-up.
> > >
> > > **Compatibility with quantization.**
> > >
> > > **Our primary goal is to unleash the capability of SNNs on commodity edge devices.** To achieve efficient deployment and broaden SNN applications, a plug-and-play framework like Spik4lite is necessary. Your follow-up feedback on compatibility with other compression methods (e.g., quantization) is very valuable. We have conducted preliminary experiments to address this.
> > >
> > > We integrated Spik4lite with Quantization Aware Training (QAT) using the Spikingformer-4-384 (T=4) architecture on CIFAR10. All experiments used the same training parameters, inference batch size (B=64), and power mode (15W) on an NVIDIA Jetson Orin Nano 8G.
> > >
> > > According to the NVIDIA Jetson Orin Series SoC Technical Reference Manual, the Jetson Orin Nano supports FP32, FP16, and INT8 precision, but it does not support FP8. Furthermore, performing INT8 inference relies on NVIDIA TensorRT, which requires converting the model to the ONNX format (NVIDIA TensorRT Documentation). However, ONNX does not support spiking neuron operators like LIF (also verified by the latest research in neuromorphic computing). Therefore, we currently cannot perform real INT8 inference on Jetson using official NVIDIA tools.
> > >
> > > Despite this, we used Fake Quantization to simulate INT8 inference. This allows us to evaluate the compatibility and efficiency gains of Spik4lite with INT8 quantization. For FP8, which lacks native hardware support, we used the same Fake Quantization method to simulate its inference performance.
> > >
> > > | Data Types | Architecture | Top-1 Acc.(%) | Latency/Batch (s) | Peak Mem. (MiB) | Param. (M) | SOPs (M) | Energy (J) |
> > > | :---: | :--- | :---: | :---: | :---: | :---: | :---: | :---: |
> > > | FP32 | Spikingformer-4-384 | 95.62 | 0.72 | 2594 | 9.32 | 444.37 | 4.39 |
> > > | | + Spik4lite | 95.14 | 0.48 | 1773 | 3.67 | 292.16 | 2.83 |
> > > | FP16 | Spikingformer-4-384 | 95.07 | 0.46 | 1389 | 9.32 | 442.85 | 2.53 |
> > > | | + Spik4lite | 94.43 | 0.29 | 878 | 3.38 | 255.34 | 1.54 |
> > > | FP8 | Spikingformer-4-384 | 94.58 | 0.85 | 2760 | 9.32 | 458.1 | 5.10 |
> > > | | + Spik4lite | 94.75 | 0.57 | 1786 | 3.69 | 292.98 | 3.31 |
> > > | INT8 | Spikingformer-4-384 | 94.57 | 0.84 | 2760 | 9.32 | 468.59 | 4.98 |
> > > | | + Spik4lite | 94.92 | 0.56 | 1779 | 3.62 | 295.68 | 3.31 |
> > >
> > > As shown in the above table, **Spik4lite achieves significant efficiency improvements across different data types with minimal performance loss.** Moreover, combining Spik4lite with FP8 or INT8 quantization does not cause further accuracy degradation. Instead, compared to using only quantization, combining with Spik4lite can even help to improve the accuracy slightly. This narrows the accuracy gap with the original FP32 baseline. The results show that Spik4lite is compatible with quantization methods.
> > >
> > > Although FP16 quantization provides some efficiency improvements on edge devices, Spik4lite further enhances these gains. The improvement brought by Spik4lite is greater than the benefit of FP16 quantization. **Spik4lite contributes to most of the efficiency for the baseline model on commodity edge devices.** Taking inference latency as an example, the total latency reduction from the FP32 baseline to the combined FP16+Spik4lite model is 0.43 seconds (0.72s to 0.29s). Of this total improvement, Spik4lite accounts for 55.8% of the reduction (0.24s), while FP16 quantization accounts for 44.2% (0.19s).
> > >
> > > Recall that our research objective is to unleash the capability of SNNs on commodity edge devices. For instance, FP8 quantization requires native hardware support, while INT8 inference is constrained by TensorRT requirements. These restrictions do not align with our vision for commodity edge devices. In contrast, **Spik4lite does not rely on specific hardware features or specific inference acceleration tools.** Thus, it offers broader applicability and more significant efficiency improvements.

---

### Decision · Program_Chairs · 2026-04-30

**Decision:**

Accept (regular)

**Comment:**

This work addresses the critical issue of inefficient deployment of SNNs on commodity edge devices by proposing the Spik4lite framework. Through an energy-aware channel gating mechanism and physical channel removal, it successfully translates theoretical sparsity into practical hardware efficiency, with a rigorous design and outstanding engineering value.The authors have responded effectively to all key concerns raised by reviewers regarding baseline comparisons, generalization, and deployment details by supplementing ImageNet experiments, quantization compatibility validation, and per-sample latency/energy measurements.The proposed approach achieves significant improvements in low latency and low power consumption on the Jetson platform, while offering plug-and-play generality, making it highly impactful for the real-world deployment of SNNs in edge computing.Overall, the paper is technically sound with comprehensive revisions and is therefore recommended for acceptance.